# Investigating the Role of Overparameterization While Solving the Pendulum with DeepONets

**Pulkit Gopalani**
Department of Electrical Engineering
IIT Kanpur, India
`gpulkit@iitk.ac.in`

**Anirbit Mukherjee**
Department of Computer Science
University of Manchester, UK
`anirbit.mukherjee@manchester.ac.uk`

## Abstract

Machine learning methods have made substantial advances in various aspects of physics. In particular multiple deep-learning methods have emerged as efficient ways of numerically solving differential equations arising commonly in physics. DeepONets [22] are one of the most prominent ideas in this theme which entails an optimization over a space of inner-products of neural nets. In this work we study the training dynamics of DeepONets for solving the pendulum to bring to light some intriguing properties of it. We demonstrate that contrary to usual expectations, test error here has its first local minima at the interpolation threshold i.e when model size ≈ training data size. Secondly, as opposed to the average end-point error, the best test error over iterations has better dependence on model size, as in it shows only a very mild double-descent. Lastly, we show evidence that triple-descent [1] is unlikely to occur for DeepONets.

## 1 Introduction

Machine learning methods have made substantial forays into various aspects of physics like steering a quantum system towards desired dynamics [5], computing invariants of Calabi-Yau manifolds [13] and knots [10]. A particularly active area in this theme has been what is called "physics informed machine learning" [15] which broadly refers to the methods used for numerically solving differential equations using neural nets. Notable recent work in this direction includes Physics Inspired Neural Nets ([28], [29], [30]), Neural Operators ([19], [18]), DeepONet ([22]), [6], [20]) Physics Inspired DeepONet ([32]) and DeepRitz Method ([33]). Experiments so far suggest the method of DeepONets ([22]) to be particularly powerful and versatile.

An intriguing aspect in neural net training is that of population risk having one or two local maxima (called "double/triple descent" respectively) w.r.t the model size and it being monotonically decreasing for nets larger than a threshold value. Recently the importance of Double-Descent was brought to light in [4], where it was observed for a number of models like decision trees, random features and two-layer neural networks. Evidence for double-descent for deep NNs like ResNets and Transformers was presented in [27], which showed epoch-wise double-descent, and also that the double-descent curves at different training set sizes can intersect in a way s.t there can be over-parameterized nets where using more training data hurts. Further evidence of double-descent appeared in, [31], [8] and evidence for triple-descent has also appeared in [1].

From all these examples, a few traits have emerged which are now believed to be universal across all supervised deep learning experiments:

- As the model size increases the minimum achievable test error can have multiple peaks but one local maxima always occurs in the vicinity the interpolation threshold i.e when the number of parameters is nearly the same as the training data-size.
- The above peak is exacerbated by the label noise.

DLDE Workshop in the 35th Conference on Neural Information Processing Systems (NeurIPS 2021).

- Asymptotically in the model size the minimum achievable test error keeps decreasing.

In light of these results we ask the following questions about training a DeepONet:

*Does DeepONet training have a double-descent phenomenon at the interpolation threshold i.e when number of training data samples = the number of trainable parameters?*

*Does DeepONet training have a triple-descent phenomenon with the peaks of the test error vs number of parameters curve happening at the two thresholds identified in [1]?*

*How are any of these peaks in the above two phenomenon affected by injecting noise into the training data (and not the test data)?*

In this work we study the properties of DeepONets from the viewpoint of understanding how the population risk (expectation of the error over the true distribution of data) of a well-trained DeepONet changes w.r.t model size, training data size and noise.

We note that in the setup of the last question above, we are not adding the extra noise in the test-data. Hence this explores the phenomenon of double/triple descent in DeepONets while at the same time testing for out-of-distribution robustness i.e DeepONet's ability to generalize on true data while being trained on data of lower quality.

## 2 The Experimental Setup

### 2.1 The DeepONet Architecture

The DeepONet $\mathcal{U}$ we use maps as, $\mathcal{U} : \mathbb{R}^m \times \mathbb{R} \to \mathbb{R}$. where $m$ is the number of "sensor points". Given an input $(\mathbf{x}_{\text{func}}, x_{\text{loc}}) \in \mathbb{R}^m \times \mathbb{R}$ to $\mathcal{U}$, the DeepONet can be described as:

1. **The branch network:** This takes in as input $\mathbf{x}_{\text{func}}$, and has $p$ feedforward, fully-connected layers with ReLU activations. The first layer has input dimension of $m$ and output dimension $w$. For the subsequent $p-1$ layers we set input dimension = output dimension = $w$.

2. **The trunk network:** For the same $p$ and $w$ as above, this takes in an input $x_{\text{loc}} \in \mathbb{R}$, and has $p$ feedforward, fully-connected layers with ReLU activations. The first layer has input dimension of $1$ and output dimension $w$ and for the subsequent $p-1$ layers we set input dimension = output dimension = $w$.

3. **The Output Layer:** This computes the inner product of the outputs of the branch net and the trunk net and adds a bias term to the result.

In our experimental study we choose $m = 100$ and $p = 4$ for all tests.

### 2.2 The Training Data and the Loss Function

For some "forcing function" $u : \mathbb{R} \to \mathbb{R}$, and "spring constant" $k$ (set to $1$ in the experiments), the O.D.E. to solve for the pendulum problem can be given as follows for a dynamical variable $s \in \mathbb{R}^2$,

$$\frac{\mathrm{d}s}{\mathrm{d}t} = \mathbf{g}(\mathrm{s}, u, t) = (s_2, -k \cdot \sin(s_1) + u)$$

In light of the above, each training/test data can be seen as a $3$–tuple, given by $(\mathbf{x}_{\text{func}}, x_{\text{loc}}, y)$, where $\mathbf{x}_{\text{func}} = u(\mathbf{x}_{\text{sensors}}) \in \mathbb{R}^m$ for $\mathbf{x}_{\text{sensors}} = (x_1, x_2, ..., x_m) \in \mathbb{R}^m$, where $\mathbf{x}_{\text{sensors}}$ is an $m$–sized equispaced grid on the $[0, 1]$ interval.

For our experiments, we sample $u$ as a random linear combination of the first 20 Chebyshev polynomials (of the first kind) by sampling coefficients uniform randomly from $[-1, 1]$.

$x_{\text{loc}} \in \mathbb{R}$ is a uniform random sample $\sim [0, 1]$. $y \in \mathbb{R}$ is the output for the above given training points. This is calculated as a standard O.D.E solver's approximate solution of the pendulum O.D.E. over an interval $[0, t_f]$ where $t_f = x_{\text{loc}}$. We refer the reader to [22] for more details about this type of data.

Hence for $\mathcal{S}$ training data samples, the $\ell_2$ empirical loss is,

$$\hat{\mathcal{L}}_{\text{DON}} (\text{DeepONet}) := \frac{1}{\mathcal{S}} \sum_{i=1}^{\mathcal{S}} (y_i - \text{DeepONet}(\mathbf{x}_{\text{func,i}}, x_{\text{loc,i}}))^2$$

### 2.3 Implementation

We train the DeepONet above on clean training data as well as when the true labels are distorted by a standard normal additive noise at standard deviations of $0.001, 0.01, 0.1, 1.0$. Note that we never add noise to the test-data and hence we are implicitly testing for robustness of the training.

We train the models using deterministic-Adam (full dataset per iteration) and stochastic-Adam (mini-batch size of 32 per iteration) [16] with a learning rate of $0.0001$, $(\beta_1, \beta_2) = (0.9, 0.999))$ on two datasets of different sizes ((#Train, #Test) = $(1000, 200)$ and $(5000, 1000)$) measuring the best error attained during the whole training and the average end-point error - defined as the error average over the last $10^4$ iterations of the training.

Our code is built on top of the work in [22], covered under the Apache-2.0 license. Our implementation is available here. We ran all experiments on Google Colab [9] using the single GPU provided therein (NVidia Tesla T4 or K80). The training required for this paper took approximately 70 GPU-hours. Data used in this paper is included in the above implementation.

## 3 Results

In Figures 1 - 4, we show the training error attained at different model sizes (along the x-axis) on clean data and the test error (on clean data) attained at the same model sizes - for clean training data as well as for different levels of noise contamination in the labels of the training data as described above.

In Figures 5 - 6 we overlay the test-error curves for both datasets (1K and 5K training samples). (Solid lines denote #Train = 1000, dashed lines denote #Train = 5000).

In any of the plots the vertical dashed lines indicate the usually benchmarked interpolation thresholds of number of trainable parameters = (training data size)$^k$ for $k = 1, 2$ as in [1], [27]. **(Full size plots are available at this link.)**

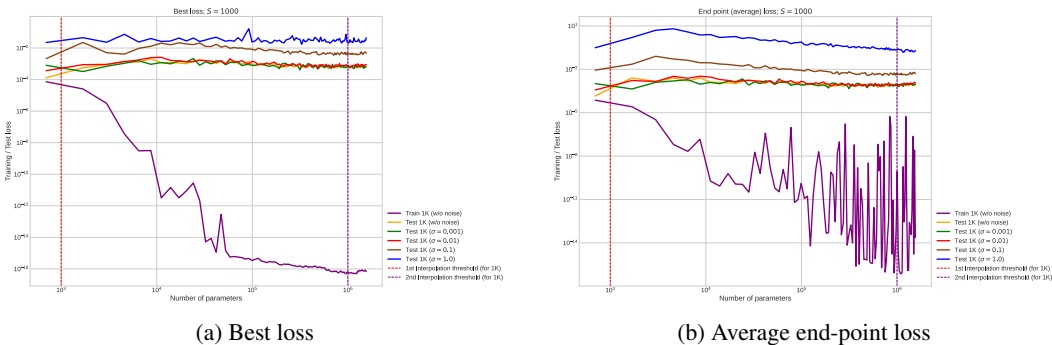

(a) Best loss        (b) Average end-point loss

Figure 1: Test and train error vs model size curves for training via deterministic-Adam using $1000$ training samples

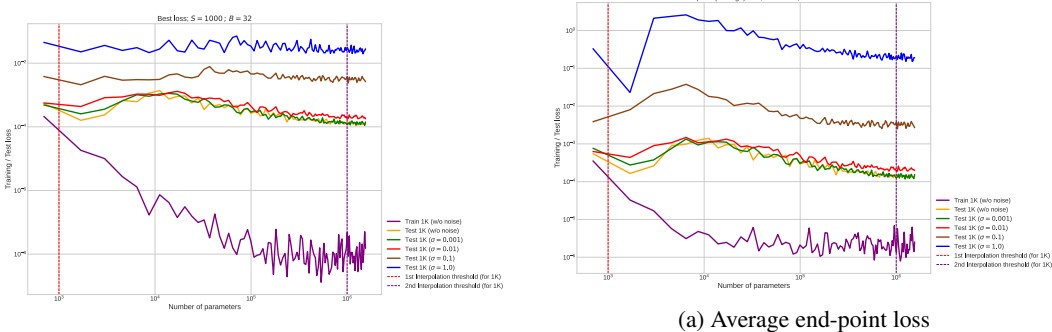

(a) Average end-point loss

Figure 2: Test and train error vs model size curves for training via stochastic-Adam using $1000$ training samples

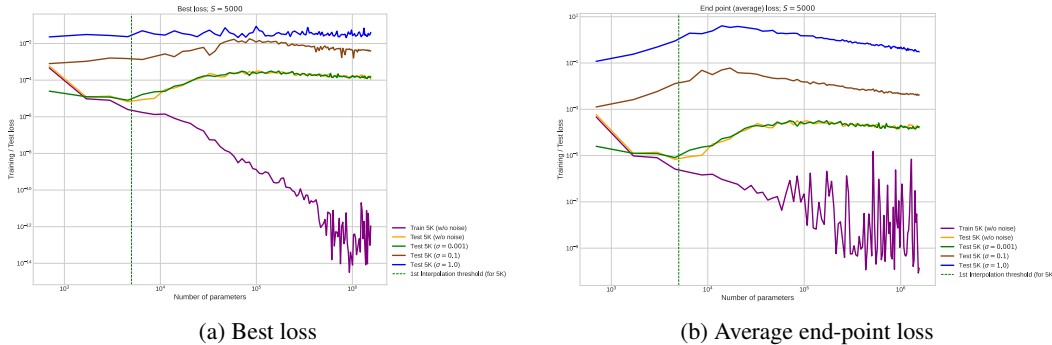

(a) Best loss

(b) Average end-point loss

Figure 3: Test and train error vs model size curves for training via deterministic-Adam using 5000 training samples

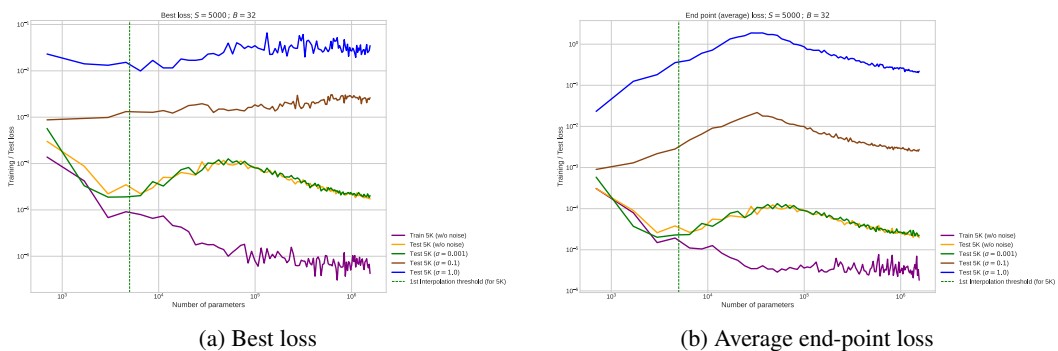

(a) Best loss

(b) Average end-point loss

Figure 4: Test and train error vs model size curves for training via stochastic-Adam using 5000 training samples

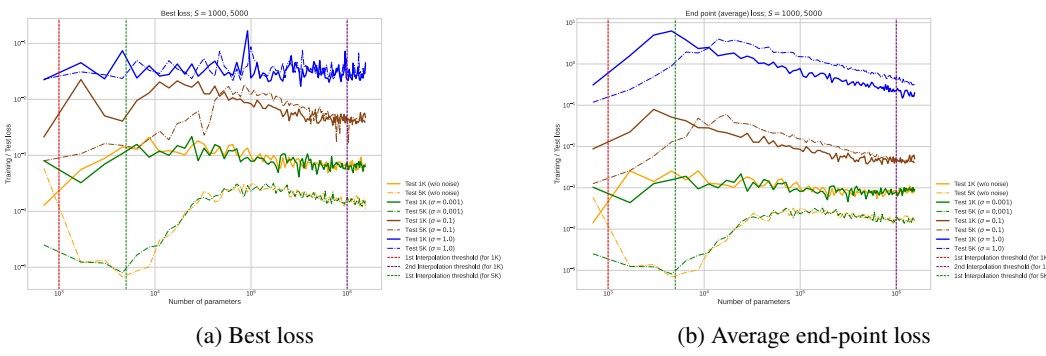

(a) Best loss

(b) Average end-point loss

Figure 5: Test error vs model size curves for training via deterministic-Adam, #Train = 5000 vs 1000

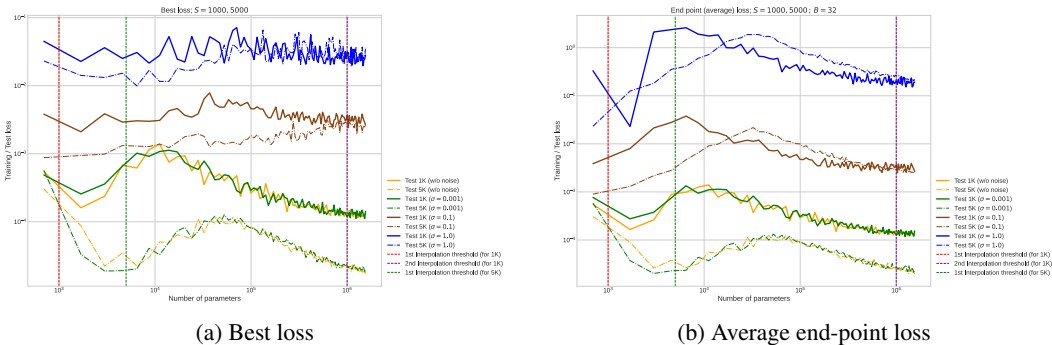

(a) Best loss  (b) Average end-point loss

Figure 6: Test error vs model size curves for training via stochastic-Adam, #Train = 5000 vs 1000

# 4 Conclusion

From the data displayed above *firstly* and *most crucially* we observe that for the cases when the training and the test data are uncorrupted, the first local minima of the best test error-vs-model size curve occurs for number of parameters equal to the training-data size (the interpolation threshold) or when slightly more than that. We note that this is contrary to the observations in [27] with standard neural nets where they observed a local maximum of test error at the interpolation threshold and a local minima to the left of it. So the "classical statistics" region possibly lasts longer for DeepONets, where increasing the model size monotonically keeps benefitting.

*Secondly,* we note that the best loss (over iterates)-vs-model size curve has significant double-descent only for training via stochastic-Adam. And that too is largely removed when the noise in the training data has high enough variance - but at the cost of overparameterization not giving an advantage in terms of getting lower test errors.

On the other hand, the average endpoint loss case always suffers from double-descent and for training via stochastic-Adam its most pronounced relative to the magnitude of the effect for the best loss.

*Thirdly,* from Figures 5 and 6, we note that like in the experiments in [27], we do see situations where having more training data hurts - but this effect is significant only for the average end-point loss and not when measuring the best loss over iterates.

Thus combining the two points above, we motivate the proposal for preferring to use the best loss over iterates as the result of training DeepONets as opposed to average loss at the end of the training.

*Lastly*, our results lead us to believe that DeepONets are unlikely to suffer from triple descent at the locations specified in [1].

# 5 Future Work

In the experimental setup (Section 2) for the Branch and Trunk net, we choose both to be of the same depth, and their intermediate layers to have the same input and output dimensions. We leave the analysis of not using such 'symmetric' DeepONets to future work. Also it remains to be seen whether the phenomenon described here holds for more complicated situations too like P.D.Es.

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
