# OpenReview forum: "Investigating the Role of Overparameterization While Solving the Pendulum with DeepONets"
_NeurIPS.cc/2021/Workshop/DLDE — DLDE Workshop -- NeurIPS 2021 Poster_

### Official Review · Reviewer_Bero · 2021-10-11

**Confidence:** 4

**Review:**

Summary:

The paper under review investigates the effects of overparametrization when learning the solution of the ODE for the pendulum problem via so-called DeepONets, which are certain neural networks. This is a simple supervised learning task where the training data is generated via a standard ODE solver. The behavior of the training and test error as a function of the number of parameters is then analyzed. The test error has a first local minimum when the number of parameters equals the training data size which is in contrast to standard observations. Moreover, no significant double descent and triple descent phenomena are observed.

Comments:

The paper provides an empirical study for a very specific setting without giving (mathematical) reasons for the observed phenomena. In my opinion the conclusions cannot be translated to general DeepONets since the architecture and in particular the problem is very specific. In particular the research questions asked in Section 1 have only be answered in the case of the pendulum problem under a very specific architecture. Moreover the results should be compared with the ones of [22] where theoretical results are provided and where  for instance the gravity pendulum is analyzed.

**Score:**

2: Borderline paper

---

### Official Review · Reviewer_LZf7 · 2021-10-11

**Confidence:** 4

**Review:**

The authors study the effects of overparameterization in DeepONets for finding the solution for the differential equation of a pendulum. They do this by turning the task of finding the solution to a supervised learning task with appropriate input and output data. Through their experiments they show that contrary to observations in the supervised learning literature, for DeepOnets for solving the pendulum, the first local minima (instead of maxima) occurs at the interpolation threshold. They further show that double descent only occurs under certain conditions and they do not observe any triple descent.

I feel that the scope of the empirical study performed in the paper is too limited for any substantial conclusions to be made, either about DeepOnets or any task that looks for solving a differential equation. As observations to an empirical study, yes the results are interesting, but the paper lacks any reasoning, mathematical or otherwise for the observations that they make.

**Score:**

2: Borderline paper

---

### Official Review · Reviewer_SMqi · 2021-10-11
**Does the double descend phenomena not apply to DeepONets?**

**Confidence:** 3

**Review:**

The authors investigate the train and test error of DeepONets as a function of the number of parameters on the pendulum problem. They empirically show that the double descent phenomena observed with e.g. Deep ResNets on MNIST/CIFAR data is not seen with DeepONets for the pendulum problem.

It would be worth demonstrating that other architectures than DeepONet show double descent on the pendulum problem. Without this it's hard to assess the value of this work. The experiments are interesting - but would be easier to judge the results with simpler graphs. There's some other notable differences from the double descent work in [27], for example adding label noise seems to shift the local maxima of test error towards models of lower complexity. I'm not sure whether these discrepancies between [27] are due to the notion of generalization for this particular task, the way data is sampled from learning differential equations or because of the choice of architecture. It would be good to see a comparison between DeepONets and for example fully-connected networks on this task.

The work lacks any discussion of why DeepONets might behave differently when effects of overparameterization are being considered. If the authors would contrast this architecture with others in their experiments, on the pendulum task, that would perhaps help further analysis.

Minor Things:

I would recommend making the figures using vector graphics, they're blurry and hard to read. I would also recommend keeping the training curve and the validation curves on separate figures. For some of the figures it's hard to clearly distinguish local maxima/minima in the error curves.


**Score:**

2: Borderline paper

---

### Decision · Program_Chairs · 2021-10-17

**Decision:**

Accept (Poster)

**Comment:**

The paper considers an interesting architecture (DeepONets) recently proposed for solving dynamical equations, and the authors investigate the double descend phenomena for this setting on the pendulum problem. The work has potential to be an interesting contribution to statistical learning theory in the context of dynamical systems applications. The authors are highly encouraged to address reviewer's remarks for their poster, such as by reviewing how the double descend phenomena emerges in the 'usual setting', on the pendulum task, when DeepONets are not applied as a comparison study that can better highlight the reported results.